# Unravelling the Potential of Salivary Volatile Metabolites in Oral Diseases. A Review

**DOI:** 10.3390/molecules25133098

**Published:** 2020-07-07

**Authors:** Jorge A. M. Pereira, Priscilla Porto-Figueira, Ravindra Taware, Pritam Sukul, Srikanth Rapole, José S. Câmara

**Affiliations:** 1CQM–Centro de Química da Madeira, Universidade da Madeira, Campus da Penteada, 9020-105 Funchal, Portugal; priscillaportofigueira@gmail.com; 2Proteomics Lab, National Centre for Cell Science (NCCS), Ganeshkhind Road, SPPU Campus, Pune 411007, India; ravindra.taware@nccs.res.in (R.T.); rsrikanth@nccs.res.in (S.R.); 3Department of Anaesthesiology and Intensive Care Medicine, Rostock Medical Breath Research Analytics and Technologies (ROMBAT), Rostock University Medical Centre, 18057 Rostock, Germany; pritam.sukul@uni-rostock.de; 4Faculdade de Ciências Exatas e da Engenharia, Universidade da Madeira, Campus da Penteada, 9020-105 Funchal, Portugal

**Keywords:** saliva, volatile organic compounds (VOCs), oral diseases (ODs), biomarkers, analytical platforms

## Abstract

Fostered by the advances in the instrumental and analytical fields, in recent years the analysis of volatile organic compounds (VOCs) has emerged as a new frontier in medical diagnostics. VOCs analysis is a non-invasive, rapid and inexpensive strategy with promising potential in clinical diagnostic procedures. Since cellular metabolism is altered by diseases, the resulting metabolic effects on VOCs may serve as biomarkers for any given pathophysiologic condition. Human VOCs are released from biomatrices such as saliva, urine, skin emanations and exhaled breath and are derived from many metabolic pathways. In this review, the potential of VOCs present in saliva will be explored as a monitoring tool for several oral diseases, including gingivitis and periodontal disease, dental caries, and oral cancer. Moreover, the analytical state-of-the-art for salivary volatomics, e.g., the most common extraction techniques along with the current challenges and future perspectives will be addressed unequivocally.

## 1. Introduction

Worldwide, oral diseases (ODs) are highly prevalent affecting over 3.5 billion people, particularly in low- and middle-income countries (LMICs) [1]. ODs compromise very seriously the overall health and wellbeing, causing a myriad of conditions to the patient, as pain, sepsis and reduced life quality. Additional side effects, as loss of school days, family disruption, decreased work productivity or dental treatment costs, have a significant impact in the healthcare systems and society. According to Peres et al. [1], dental caries (tooth decay), periodontal disease, tooth loss, and cancers of the lips and oral cavity are among the most prevalent ODs. Although different procedures have been described for early detection in order to cope with these diseases, the diagnostic efficiencies greatly depend on the clinical conditions of the patients [2,3,4]. Nevertheless, a non-invasive method using saliva for ODs diagnostics is a highly attractive strategy. Saliva is often described as the “mirror of the body” [5,6] and so different omics approaches such us proteomics, metabolomics and more recently, volatomics are being employed to explore the potential of this biofluid towards the non-invasive diagnosis of ODs [5].

## 2. Physiology of Saliva

### Saliva Composition and Production

In its broader sense, saliva composition includes the fluids produced by the different salivary glands and the gingival-crevicular secretions, along with cellular components, oral microbiota as well as food debris [7]. Saliva is a colourless liquid, slightly acidic, hypotonic and a mucoserous exocrine biological medium mainly composed of water (98%) with a production of approximately 0.5–1.5 L per day [5,6,8]. It is also possible to find different organic compounds including hormones, peptides, mucus, antibacterial compounds, enzymes, proteins, urea, uric acid, lactate and creatinine [2,4,8,9]. Ions such as Na^+^, K^+^, Ca^2+^, Cl^−^ and HCO^3−^, are also present in saliva [5,9]. The overall composition of saliva is affected by internal (related to the type of secretory glands and their stimulation) and external (personal habits including diet and lifestyle) factors [5,9,10]. A deeper understanding of the interplay of these factors on saliva production and composition can be found elsewhere [11,12].

The autonomic nervous system plays a key role in the production of saliva. This control is exerted by various nerves that regulate the viscosity, flow rate, volume and concentration of various constituents of saliva. The sublingual and minor mucus glands are regulated by stimulation of sympathetic nerves whereas the parotid and submandibular glands are stimulated by the parasympathetic innervation [4,5,8,13]. Saliva is mostly produced by three pairs of major salivary glands: i) parotid (which contribute for ~23% of a total saliva production), ii) submandibular (which contribute for ~65% of a total saliva production) and iii) sublingual (which contribute for a ~4% of a total saliva production (Figure 1) [4,5,8,9]. Additionally, numerous minor salivary glands (around 300–1000 units of von Ebner’s, mucous gland, labial and palatine glands, among other) distributed throughout the oral mucosa contribute with around ~8% of total saliva production [4,5,8,13]. Each type of salivary gland produces a specific secretion [8,10]. The parotid produces a serous fluid, the submandibular a sero-mucous, the sublingual secretes only mucous saliva and the minor glands produce viscous secretions [8,10]. The initial or primary saliva is an isotonic fluid produced by the acini, a cluster of cells (acinus) with a berry-type structure that are present in the salivary glands. These acini drain into the salivary ducts with small ‘striated’ ducts opening into wider intercalated and excretory ducts [10,14]. The acini are surrounded by blood capillaries that enable the exchange of substances from the circulation into the salivary glands [10]. This transport is done through three distinct mechanisms, namely selective transport using passive diffusion, ultrafiltration through pores or active (energy-dependent) transport against concentration gradients leading to the enrichment of the saliva composition [8,10,14,15] (Figure 1). During the passive diffusion, several compounds cross the cell membranes of capillaries and acini. Lipid-soluble compounds (e.g.,: unconjugated steroids) are close to unbound plasma concentrations, while hydrophilic compounds (e.g.,: conjugated steroids) reach only 1% of unbound plasma concentrations [3,8,14]. In turn, small polar molecules (having molecular weight below 1.9 kDa, as ions and some hormones) can be transported to saliva by ultrafiltration [8,14]. This transport is done through narrow junctions in the apical pole of the acini. Finally, for some electrolytes and larger molecules like peptides, the passage from capillaries to saliva will be dependent on an active transport mechanism [8,10,14]. Beyond these mechanisms of transport from the capillaries, elements such as bacteria, epithelial cells, erythrocytes, leukocytes, food debris or other contaminants may be also present in the saliva, causing significant variations to the final saliva composition [3,8]. This fact is particularly relevant in the context of the applicability of saliva as a diagnostic tool for different diseases. The connection between local (salivary glands) and systemic (blood) circulation sources provides a good option for looking into disease biomarkers or studying a physiological state. Moreover, saliva can be collected non-invasively, more affordable, accessible and painless way than blood, for instance [8,13].

## 3. Putative Salivary Biomarkers for Oral Diseases

As mentioned, ODs are a serious problem worldwide, affecting mostly the poor and marginalized groups in society with particularly higher incidence rate observed in the LMICs [1]. Strikingly, the most prevalent ODs tooth decay (caused by dental caries) and periodontal disease (gum inflammation) could be easily preventable with better oral care and access to medical services. It is therefore remarkable how such problems continue to be neglected. Cancers of the lips and oral cavity are also very prevalent, ranking in the top 15 of most common cancers in the world in 2018 [17]. Several other conditions can affect the oral cavity and its soft and hard tissues, but fortunately they have a lower global impact. The risk factors for the development of ODs are well-known, being the most relevant the abuse of sugar, alcohol and tobacco. In turn, these risk factors are also the main cause of other clinical conditions, as diabesity (diabetes and obesity co-occurrence), caries (excessive sugar consumption) and different forms of cancer (alcohol and tobacco abuse) [1]. Therefore, it is not surprising that great deal of efforts were made to identify the salivary biomarkers for ODs.

Despite its apparent simplicity, saliva contains a large repertoire of proteins (>3600) and peptides (>12,000), being a great part of them (~50% of proteins and 80% of peptides) also found in plasma [18,19]. This feature is most probably caused by the exchange of molecules with the crevicular fluid at the periodontal pocket of sulcus through passive diffusion, pinocytosis and the fusion pores of acinar cells as represented in Figure 1 [20,21]. Consequently, saliva can be a very informative biofluid regarding the hormonal, immunological, nutritional and physiological status of an individual [7]. Most of the proteins found in saliva are low molecular weight proline-rich proteins transcribed from chromosome 12 [22]. The salivary proteome is largely modified by various post-translational modifications including glycosylation, phosphorylation, acetylation, ubiquitination, and methylation among others which constitute a genome-independent and specific “biosignature” [23]. Such signature will be obviously modulated by specific patterns of proteome alterations associated with ODs, constituting an important diagnostic tool to unveil putative biomarkers for different conditions. At this point should be clarified that many of the studies included in this section points to putative biomarkers that still require further validation before they can be reliably used in the clinical environment.

### 3.1. Gingivitis and Periodontal Disease

Gingivitis constitutes the initial stage of periodontal disease characterized by the inflammation of gums or gingiva. It is caused by the bacterial plaques build up on teeth and consequent inflammation of the surrounding tissues [24]. If not treated properly, gingivitis will progress to periodontal disease-causing tissue and bone degradation and eventual tooth loss. A non-targeted proteomic analysis of saliva by two-dimensional gel electrophoresis (2DE) and matrix-assisted laser desorption/ionization time-of-flight mass spectrometry (MALDI-TOF MS) identified the upregulation of immunoglobulin and keratins in gingivitis patients (*n* = 10) as compared to controls (n = 10) [25]. Another non-targeted study of saliva, found several proteins (among mucins, albumins, lactoferrins, histatins and amylases) altered in periodontitis [26,27,28,29]. The label free quantitative proteomic analysis of a periodontitis cohort by Bostanci et al. [30] revealed that lactoferrin, lacritin, sCD14, Mucin 5B and Mucin7 levels were reduced as compared to controls. This result suggests that the lower disease resistance offered by periodontitis patients is due to the reduced antimicrobial properties exhibited by their saliva. The matrix metalloproteinases (MMPs) are the proteases involved in the extracellular matrix remodelling and linked to periodontal inflammation and collagen degradation [31]. Particularly, MMP8 is considered as important biomarker for periodontitis. Furthermore, recent proteomic investigation also confirmed that MMP8 along with other cytokines (interleukin-1b–IL-1b, RANK/RANKL/OPG) were overexpressed in periodontitis [32]. In fact, Gursoy et al. [33] proposed a cumulative risk score using the salivary concentrations of *Porphyromonas gingivalis*, IL-1β, and MM8 as a novel and non-invasive model for the risk assessment of advanced periodontitis. The level of several other salivary cytokines, IL-6, IL-8, IL-17A, and tumour necrosis factor α (TNF- α) have been shown to be affected by periodontitis progress [34]. The identification of so many inflammatory markers in different studies involving periodontitis seems obvious given the progression of the gum inflammation from an initial gingivitis stage to periodontitis. In this case, however further studies are necessary to identify which markers are specific to the gum inflammation.

Aimetti et al. [35] carried out metabolomic analysis of saliva samples applying proton nuclear magnetic resonance (^1^H-NMR) to a cohort of 43 individuals (21 chronic periodontitis patients and 22 controls). They found that high concentrations of short chain fatty acids such as acetate, propionate, *n*-butyrate along with succinate, valine, phenylalanine and trimethylamine are associated with chronic periodontitis. In contrast, the authors also detected low pyruvate and *n*-acetyl groups concentrations in same study cohort. Several other studies using the same approach and clinical cohorts of periodontitis of various strengths also identified augmented levels of amino acids, alcohols, short chain fatty acids and sugars in the disease group [36,37,38,39]. Apart from ^1^H-NMR, gas chromatography-mass spectrometry (GC-MS) and liquid chromatography-mass spectrometry (LC-MS) were also employed to detect metabolic signatures associated with the periodontitis. Liebsch et al. performed ultra-high pressure liquid chromatography-tandem mass spectrometry (UHPLC-MS/MS) metabolomic analysis of a cohort of 909 subjects and identified the bacterial metabolite phenylacetate as a potential biomarker for periodontitis across all age groups [40]. Huang et al. carried out targeted (LC-MS) and untargeted (GC-MS) lipid profiling of saliva samples in a study involving 50 periodontitis patients and 50 healthy controls. The authors observed increased levels of some prostaglandins, namely PGE2, PGD2, PGF2α, and thromboxane 2, and decreased levels of others, such as PGI2, in periodontitis patients. Moreover, they also reported elevated levels of 5-HETE (arachidonic acid metabolite) and decreased levels of 13-HODE and 9-HODE (linoleic acid metabolites) in periodontitis patients [41]. In another study, the salivary macrophage inflammatory protein-1α, MM8, IL-1β, IL-6, prostaglandin E2 and TNF-α levels were reported to correlate with gingivitis and periodontitis development [42]. Finally, aggressive periodontitis has been associated to an exacerbated production of adrenomedullin and nitric oxide (NO) in saliva [43]. At a different level, the detection of *P. gingivalis* using saliva kits may constitute another important tool for ODs study by providing an easy and time-efficient chair-side tool for the detection of *P. gingivalis* [44].

### 3.2. Dental Caries

Tooth decay, also known as dental caries, is one of the major ODs observed across all age groups worldwide. This multifactorial disease is triggered by the excessive consumption of fermentable carbohydrates (sugars, as glucose, fructose, sucrose and maltose), poor oral hygiene and inadequate fluoride exposure [45]. This combination eventually results in the formation and progression of a microbial biofilm producing acids which cause the demineralization of dental tissues and finally dental cavities [45]. Given that the prevalence of dental caries is positively correlated with the microbial load of *Streptococcus mutans* and *Lactobacillus* in the saliva [46], these two bacteria should be involved in the cascade leading to tooth decay. *Porphyromonas gingivalis* is another bacterium with a relevant role in ODs as it was implicated in the biofilm formation of bacterial plaque. Furthermore, it also plays an important role in the progression of periodontal disease as well as in the onset of different systemic pathologies, including rheumatoid arthritis, cardiovascular pathologies, and neurodegenerative pathologies (reviewed in [47]).

Vitorino et al. [45] used a gel-based proteomic approach to study dental caries formation and identified reduced levels of acidic proline rich phosphoproteins associated with the higher risk of caries [48]. However, the other proteins involved in this study, such as agglutinins, amylase, lactoferrin, lysozyme and some antibacterial peptides, were found to be inconsistent with a diagnostic potential. In turn, Fidalgo et al. [46] carried out ^1^H-NMR studies of saliva sample to identify the metabolomic fingerprint associated with the dental caries in children [49]. In a study group of 33 subjects (dental caries = 15, control = 18), the authors identified lactate, acetate and n-butyrate significantly increased in dental caries subjects as compared to control group.

### 3.3. Oral Cancer

Oral cancer is the sixth most frequent malignant disease across the globe and oral squamous cell carcinoma (OSCC) is the most often reported subtype. The onset of oral cancer is often asymptomatic, but overall seems to be outcome of progressive premalignant conditions such as leukoplakia and oral lichen planus [50,51]. Metabolic reprogramming in oral cancer is not yet well understood and therefore, investigation of metabolic alterations is crucial for detecting novel diagnostic biomarkers and understand the disease progression. In a couple of studies saliva was used to unveil the metabolomic signature of oral cancer. Sugimoto et al. [52] carried out a salivary metabolomic analysis by capillary electrophoresis time-of-flight mass spectrometry (CE-TOF-MS) using a cohort of 156 individuals comprising of 69 patients with oral cancer and 87 controls. Overall, 57 metabolites were detected and out of these, 28 were found to be differentially expressed in oral cancer as compared to controls. Higher levels of salivary polyamines, piperidines and taurine were detected in oral cancer group and considered as promising screening biomarkers. In another study, Ishikawa et al. [50] reported that s-adenosylmethinine and pipecolate were upregulated in the saliva of oral cancer patients as compared to the control samples and possess a good diagnostic potential for the early detection of the disease [53]. In another study, the same authors proposed that the salivary metabolites indole-3-acetate and ethanolamine phosphate have the potential to discriminate OSCC from oral leukoplakia (OLP) [54]. Wei et al. [55] performed a salivary metabolomic study in three groups comprising oral cancer, oral leukoplakia and controls by UPLC coupled with Quadrupole TOF-MS. The study revealed that the metabolic set composed by valine, phenylalanine and lactic acid exhibits a moderate specificity and sensitivity to discriminate oral cancer from oral leukoplakia and controls [55]. Using the same approach, Wang et al. [56] performed a salivary metabolomic analysis using a cohort of 60 subjects (30 oral cancer patients and 30 controls subjects). The authors reported a pool of five salivary metabolites (propionylcholine, N-acetyl-L-phenylalanine, sphinganine, phytosphingosine, and S-carboxymethyl-L-cysteine) exhibiting good sensitivity, specificity and accuracy to discriminate oral cancer patients from controls. Furthermore, Lohavanichbutr et al. [57] applied ^1^H-NMR and LC-MS/MS to unveil putative salivary metabolic signatures for oral cancer and observed that glycine and proline were significantly altered in the malignant group when compared to the controls.

The earliest salivary proteomics analysis of OSCC pointed out the role of higher levels of several cytokines (IL-6, IL-8, IL-1 and TNF-α) in the proinflammatory and proangiogenic functions [58]. Related with this, more recently Aziz et al. reported augmented levels of IL-4, IL-10, IL-13 and IL-1RA in the saliva of patients with OSCC [59]. In another study, immunoglobulins were detected at higher levels in OSCC patients as compared to controls and found to be involved in angiogenesis [60]. A similar observation was reported for several cell surface receptor glycoproteins, namely CD44, CD59 and CEA, which were found overexpressed in patients with OSCC [61]. Different zinc finger proteins were also found augmented in OSCC patients, namely ZNF510 [62] or Cyfra 21–1 and CK 19 [61,62,63]. Hu et al. [59] performed a global proteomic analysis of salivary samples (OSCC = 64, control = 64) and proposed a panel of five proteins (M2BP, MRP14, CD59, catalase and profilin) which exhibit with high sensitivity (90%) and specificity (83%) to discriminate OSCC patients from controls. In a larger study, Yu et al. carried out proteomic analysis of a cohort comprising 131 OSCC patients, 103 low risk OPMDs (oral potentially malignant disorders) patients, 130 high risk OPMDs patients and 96 controls. The authors proposed four proteins–MMP1, KNG1, ANXA2 and HSPA5 that were not only able to discriminate the OSCC from controls, but also predict OPMDs progression into OSCC [64]. In a very recent study of OSCC salivary proteome, Sivdasan et al. reported CD44, S100A7 and S100P as novel potential biomarkers for the early detection of OSCC [65]. DNA methylation and conformational alterations on histones are potential molecular signatures for different cancers, including those affecting the salivary glands [66]. Under this context, Li et al. observed that the long non-coding RNA (lncRNA) RBM5-AS1, which promotes the proliferation, migration, and invasion of OSCC cells in vitro, was highly expressed in OSCC tumour tissues and cancer cell lines [67]. Furthermore, Park et al., reported that OSCC patients have lower levels of salivary miR-125a and miR-200a, strongly suggesting that specific salivary miRNAs can be used in oral cancer detection [68].

### 3.4. Oral Potentially Malignant Disorders (OPMD)

OPMDs are clinical manifestations that aggregate conditions as the oral lichen planus and oral leukoplakia (OLP), which possess risk of malignant transformation. Several prospective studies have predicted a 1% progression rate of OPMD into oral cancer [69,70]. Yang et al. [71] carried out 2DE and MALDI-TOF MS analysis of an OLP study cohort comprising 20 cases (patients = 10, control = 10) and reported an urinary up-regulated prokallikrein and reduced palate, lung and nasal epithelium carcinoma-associated protein (PLUNC) as novel biomarkers for oral lichen planus. In another study, Souza et al. [72] discovered a positive correlation of S1008, S1009 and haptoglobin with the inflammatory cytokines and oral lichen planus pathology. Furthermore, Camisasca et al. [73] performed a 2DE-MALDI-TOF analysis of a leukoplakia cohort consisting of 15 patients and 10 controls and identified keratin 10 as an important candidate protein biomarker for the disease pathology.

### 3.5. Burning Mouth Syndrome

Burning mouth syndrome is characterized by a chronic intraoral inflammation without any visible lesions in oral cavity. This syndrome is particularly prevalent in post-menopausal women and its etiology is poorly understood [74]. Ji et al. [75] reported a salivary global quantitative proteomic study to identify candidate proteins biomarkers for burning mouth syndrome in a cohort of 19 patients and 19 healthy control volunteers. The study identified three overexpressed proteins viz. α-enolase, IL-18 and kallikrein-13, as potential markers for the burning mouth syndrome.

### 3.6. Recurrent Aphthous Ulceration (RAS)

RAS is a common oral pathology recognized by ulcers in the mucosal lining of lip, tongue and soft palate. Sometimes, it also occurs at places as the isometric mucosa of the hard palate. Li et al. [76] identified dysregulated tryptophan and steroid hormone metabolism as a signature of the RAS in study cohort of 94 individuals (RAS = 45, Control = 49) by employing LC-MS/MS. This dysregulation of tryptophan metabolism and the hormonal imbalance are plausible causes for the depression, stress and reduced salivary immunity among RAS patients.

## 4. Salivary Volatomics

In the previous section, a significant number of studies identifying potential biomarkers for ODs were discussed. That included mostly higher molecular-weight constituents such as proteins. This section will focus specifically on the subset of potential biomarkers for ODs that are the volatile organic compounds (VOCs).

Saliva as most biofluids, also contains VOCs in its composition. Currently VOCs are being explored as potential biomarkers for ODs. Furthermore, many of them have been also detected in other biofluids [77,78,79,80,81], a fact that it is very important because it adds relevant information to our understanding of human metabolism in health and disease. This has obviously applications to a myriad of clinical conditions, as malignancies, infections, cardiovascular problems or genetic disorders (reviewed in Malathi, et al. [82]). This review will focus only the ODs.

According to the most recent data reviewed by Milanowski, Pomastowski, Ligor and Buszewski [78] almost 500 VOCs have been identified in the oral cavity. This includes data from reference studies as the human volatilome showing that 359 out of 1840 VOCs were identified in the saliva [79] and the 317 VOCs identified in a 10-days follow up salivary analysis [80]. In another study, 90 out of 166 VOCs were found common to saliva and axillary sweat samples [81]. Overall, salivary VOCs derive from different sources; a part of them results from the metabolic activity of different cells in the body (review in [78]) that eventually reach the serum, blood, gingival exudate, nasal cavity or the gastrointestinal reflux, among others. The acinar cells that compose the salivary glands for instance, are highly vascularized allowing the exchange of blood components including VOCs through different mechanisms as passive diffusion, ultrafiltration and active diffusion [16,88,89]. However, the oral microbiota activity as well as food debris, commercial products (toothpaste for instance) and environmental contaminants have a significant contribution to the VOCs identified in saliva. The oral cavity seems to be colonized by a huge number of bacteria (50 to 100 billion bacteria from 300 to 700 different species [81,90,91]). In fact, there are many salivary VOCs such as aliphatic amines, branched chain fatty acids, 2,3-butanedione, 2,3-pentanedione, pyrrole, indole, phenol, and volatile Sulphur-containing compounds (VSCs, as hydrogen sulphide, methyl mercaptan, dimethyl sulphide and dimethyl disulphide) that are produced by oral bacteria [78,80,81,91]. In turn, the hydrocarbons which are consistently the most abundant salivary VOCs reflect their possible origin in food, fragrances and cosmetics. Finally, many of the long-chain alkane derivatives such as hexane, octane and undecane are probably environmental contaminants (air pollutants). The same applies to aromatic compounds, as benzene, toluene, xylenes and styrene [78]. This metabolic characterization of the source of the VOCs identified in saliva (synthetized in Table 1) is crucial for diagnostic purposes. In the context of the ODs, it is of paramount importance to know the source of a given VOC so we can infer if certain clinical condition is present and retain information about its progression or response to treatment. The identification of microorganisms using their volatile metabolic activity has been already reported for several clinical conditions, including pulmonary bacterial infections caused by *Escherichia coli*, *Pseudomonas aeruginosa*, *Staphylococcus aureus* and *Klebsiella pneumoniae* [92] as well as for *Candida fungus* causing oral candidiasis [86] (Table 1). Unique VOCs signatures for bacteria species that often colonize our mouth, *S. mutans*, *L. salivarius* and *P. acidifaciens* ([93]) have been also described. These in vitro studies clearly point to the potential of salivary VOCs as putative biomarkers for ODs. In fact, several features have been associated with periodontal disease, as increased amounts of VSCs and presence of pyridines, which are absent in the saliva of healthy controls [78]. These are, nevertheless, VOCs that can arise from different pathways and sources and it is likely that many confounding factors are affecting the reported results. Also, dimethyl disulphide has been associated to halitosis and different metabolites such as indole and skatole or phenol and p-cresol have been shown to be specific of the bacterial fermentation of tryptophan and phenolic amino acids, respectively (Table 1, reviewed in [78]). There is an obvious interplay between periodontal disease and halitosis because patients affected by periodontal disease are at higher risk for halitosis detection than healthy individuals. In fact, it has been reported that the posterior portion of the tongue dorsum constitutes an important source of odorous compounds, possibly produced by *P. gingivalis* identified in these patients [94]. Nevertheless, careful analysis of such data should be taken because alternative sources such as diet, environmental contamination or even other endogenous contributions may drive variations in some of these VOCs. Blood borne halitosis for instance, is caused by malodourous compounds generated elsewhere in the organism, carried through the blood stream to the lungs where they diffuse across the pulmonary alveolar membrane to enter the breath. Regarding this, Torsten, Gomez-Moreno and Aguilar-Salvatierra [83] reported that the metabolism of at least nine medications can release dimethyl sulphide, carbon disulphide and several VSCs in breath causing a drug-related halitosis. Moreover, the metabolism of penicillamine raises the pH level favouring the proliferation of gram-negative bacteria in the oral cavity which in turn causes halitosis. Therefore, these forms of halitosis are not caused by any specific disease in the oral cavity. Anxiety is another condition that can favour changes in oral microbiota leading to significant variations in oral VSCs and consequently halitosis [84].

Cancer development and progression causes metabolic shifts that can be detected in different biofluids, including in saliva [52]. In this regard, volatomic fingerprints able to discriminate BC from healthy individuals have been already proposed [95,96]. Taware et al. applied the same principle to oral cancer and analysed the salivary volatile composition of a small cohort of 32 oral cancer patients and 27 controls. The authors reported that the volatiles 1,4-dichlorobenzene, 1,2-decanediol, 2,5-bis1,1-dimethylethylphenol and E-3-decen-2-ol were significantly associated with the malignant disease, possessing an excellent discriminatory potential [87].

## 5. Analytical Platforms Used in the Volatomic Analysis of Saliva

### 5.1. GC-MS

Gas chromatography hyphenated with mass spectrometry (GC-MS) is one of most popular analytical platforms for volatile metabolite profiling and considered as the “gold standard” of metabolomic research. As the name suggests, in this methodology the volatile metabolites are loaded in the GC column and the separated metabolites then detected by MS. The sample volatiles can be preconcentrated using various approaches such as Solid Phase Micro Extraction (SPME) or Needle Trap Micro Extraction (NTME), desorbed in the GC, separated and detected [87,95,97]. The samples that are not volatile under normal conditions can be derivatized to make the metabolites thermostable and gaseous at higher temperature.

#### 5.1.1. Solid Phase Micro Extraction (SPME)

SPME is the most popular extraction methodology for the analysis of VOCs. A polymer coated fused silica fibre is exposed to the headspace of the sample vial containing the biospecimen, which can be solid or liquid. The SPME fibres are available in various thicknesses (65 µm, 75 µm, 85 µm, 100 µm) and polymer coatings (polydimethylsiloxane, polyacrylate, divinylbenzene, carboxen and binary and ternary mixtures of these coatings). Often, to broaden the metabolic coverage polymer coats differing in their adsorption properties such as carboxen/polydimethylsiloxane or polydimethylsiloxane/divinylbenzene, can be combined in a single fibre. The VOCs accumulated in the headspace of the vial adsorb on to the SPME fibre and this interaction is dependent on the equilibrium between the sample headspace and the stationary phase. Sample conditions can be modulated to achieve a faster equilibrium by enhancing the volatile mass transfer from bulk solution to the headspace. This can be achieved by optimizing several parameters such as pH, temperature, time, agitation speed, ionic strength and sample amount [98,99]. Following VOCs extraction, the adsorbed/absorbed metabolites are thermally desorbed in the GC inlet that is maintained at higher temperature to favour the VOCs release into the injection chamber and transferred to the capillary column by high purity carrier gas such as helium. The capillary columns are made up of various materials to selectively retain VOCs of different polarity. The VOCs retained on the column elute at specific temperature gradient and are carried to the MS system for compound identification. There is however, an important constrain in the use of SPME which refers to sample storage. SPME is not particularly tailored to store the trapped VOCs for long periods and samples should be analysed readily after extraction. Therefore, when analysing large sample cohorts using automatic extraction systems coupled to the following analytical platforms, the use of a rigorous quality control of quantification will be essential to mitigate batch to batch variations. A deeper review of such strategies can be inferred from the work of Saigusa, et al. [100] devoted to the concept of quality control of multiple batches for LC-MS-based quantification. Overall, the SPME approach is easy to use, sensitive (limits of detections in ppb range), reproducible and conserve the integrity of the biological sample [101]. Therefore, it’s not surprising to find that SPME is widely used in volatomic analysis of various biospecimens [99,101,102,103,104]. An overview about the typical experimental layout for the analysis of salivary VOCs using SPME-GC/MS is provided in Figure 2. Further variations to this scheme are discussed in the following sections and refer to different microextractions approaches and analytical configurations. The goal will be the identification and validation of salivary VOCs as biomarkers for a given condition and subsequent design of POCT devices able to provide fast, reliable and near real-time analysis of such biomarkers in the clinical environment.

#### 5.1.2. Needle Trap Micro Extraction (NTME)

NTME constitutes an improvement over SPME in which the adsorbent polymer is packed inside a needle [105]. As in SPME, the adsorbent polymer of various affinities (polar, semi polar or non-polar) can be layered sequentially to increase the VOCs coverage. However, the NTME approach offers added advantages such as enhanced sensitivity and storage of sampled metabolites in the same device [97,105,106]. Experimentally, the VOCs are sampled by passing headspace air through the sorbent packed needle using a disposable syringe of appropriate volume. The remaining steps of chromatographic separation and MS detection of VOCs are similar to the SPME. NTME will certainly be very helpful for many applications, particularly those involving field sampling.

#### 5.1.3. Thin Film Micro Extraction (TFME)

TFME employs a sheet of adsorbent polymer for extraction of VOCs from relevant biospecimens [107]. This format allows a higher extraction efficiency due to the larger surface area available for adsorption. Consequently, an enhanced VOCs preconcentration within a shorter time span can be obtained [108]. TFME, however requires a modified desorption chamber to desorb and transfer the trapped VOCs to the GC column.

#### 5.1.4. Stir Bar Sorptive Extraction (SBSE)

SBSE is a sample enrichment sortion-based technique involving the use of a magnetic device under the floating sampling technology concept [109,110]. This enables the direct microextraction of a myriad of compounds from almost all type of matrices [110], including VOCs and semi-VOCs from saliva [81,111]. Furthermore, the membrane that covers the floating device is totally customizable to match the properties of the target analytes to extract. For this reason, there is nowadays a wide range of variations to SBSE, although generically using the same operation concepts of a magnetic-driven floating device.

### 5.2. Direct Injection Mass Spectrometry

Direct injection mass spectrometry has been also employed in the detection of low abundant VOCs from different biospecimens. The analytical platforms that allow this approach are mainly Proton Transfer Reaction-Mass Spectrometry (PTR-MS), Selected Ion Flow Tube-Mass Spectrometry (SIFT-MS) and Secondary Electrospray Ionization-Mass Spectrometry (SESI-MS) [112,113,114,115]. The PTR-MS is based on the generation of H_3_O^+^ ion from high purity water source which subsequently ionize the target analytes by transferring protons. Then, the ionized metabolite can be detected by various MS analysers such as TOF or quadrupole.

However, the H_3_O^+^ soft-ionization mode enables ionization of only metabolites with higher proton affinity than water, the recent advances in PTR-MS and in SIFT-MS also offer the switching-reagent-ion mode. This produces several precursor ions (H_3_O^+^, NO^+^, or O_2_^+^) for chemical ionization of the wider variety of target analytes. Hence, a higher metabolic coverage is obtained. The advantage of direct-MS is that they perform real-time quantitative analysis of metabolites without any sample preparation steps that allows online assessment of in vivo physiology, metabolism and pathophysiology [116,117,118], without the time-consuming chromatographic separation steps. In turn, SESI-MS is a recently developed technique for measurements of large molecules (e.g., those appear in breath condensates) and is eventually gaining attention in metabolomic studies.

### 5.3. eNOSE

eNOSE or electronic nose is a targeted volatomic approach for artificial olfaction and often performed by various platforms such as laser spectroscopy and chemical and nano-optical sensors [68]. This technology offers several advantages over conventional approaches in Table 2 such as easy to use and transport, requires semi-skilled technical expertise, low operational costs and real time analysis [119,120]. Moreover, eNOSE can also be customized by integrating a gas sensing arrays of choice to suit a given application. However, this technology also suffers from several drawbacks such as low sensitivity and specificity as compared to most analytical platforms. Susceptibility to moisture interference and incapability to identify single VOCs are also important disadvantages [121].

## 6. Data Analysis

The main objective of measuring different types of analytes in saliva samples is the identification of a specific pool that, at a defined concentration shows a high level of significance as putative biomarkers for the non-invasive diagnoses of different diseases [126,127]. However, to characterize this pool of analytes it is necessary to use different statistical tools to find biopatterns able to differentiate between patients and healthy individuals. The definition of the most suitable and meaningful statistical tool is very challenging. In fact, a less accurate choice can lead to extreme and undesirable diagnostic situations involving a high level of false positives and negatives. Beyond this, it is critical to have large enough cohorts to cope with the sample complexity. Too many times, the number of samples analysed is very short in comparison to the number of metabolites identified, jeopardizing the subsequent statistical analysis. Overall, it is necessary to consider a set of factors before choosing the ideal statistical tool, namely the complexity and size of the sample under study, the number of analytes and the variability within each sample set [128]. Furthermore, it is necessary to normalize the data obtained in each group to avoid errors associated with the raw data [126,128]. Traditionally, many researchers use the *p*-value as the main statistical tool in order to identify analytes whose variation is statistically significant [129,130,131,132]. However, this approach is extremely limited and often results in numerous false positives and negatives [133,134]. For this reason, multivariate statistical analysis (MVSA) became very popular to identify set of analytes that can be considered biomarkers of different pathologies [126,127,128,135]. The MVSA can be divided into two distinct categories: the unsupervised analysis, typically performed after the normalization of the data obtained; and supervised analysis, involving a deeper characterization of the relationship between variables (analytes) and cases (sample) [36,49,128,136,137].

Principal component analysis (PCA) is the most common method used in unsupervised analysis [49,128,130,131,137]. This approach reduces the number of associated factors by showing clusters according to the separation between groups. However, the disadvantage is that it does not show which variables are directly responsible for the formation of these clusters. Thus, and although it is possible to visualize the separation between groups, unsupervised techniques, such as the PCA, do not allow the creation of a classification model for the different variables [35,128]. For this reason, supervised methods become more advantageous, allowing researchers to establish important links between the sample and the variables [35,128]. According to the complexity of the sample and the final objective of the analysis, it is possible to use linear methods, such as linear discriminant analysis (LDA), whose objective is to determine a linear function based on the analyte matrix obtained and try to establish differences between the studied groups [128]. This approach is considered relatively quick and simple and is typically used to classify variables after PCA analysis. There are several LDA methods for the analysis of biological compounds, such as the discriminant analysis of partial least squares (PLS-DA) [35,49,137]. Due to its ability to distinguish between groups, identifying which compounds are responsible for each separation, PLS-DA is an extremely useful tool in understanding the metabolites that are responsible for the separation between the control and the patients groups under study [35,49,126,128,131,136,137,138,139]. However, it is important to note that for the success of this method, the PLS-DA must be optimized a priori [140], with its linear model being trained so that its precision is high and the errors evaluated [126,128,139]. On the other hand, metabolites may not respond linearly to a mathematical model [128], and this may limit the feasibility of the linear methods previously described. To overcome this issue, another supervised approach, called non-linear analysis, was proposed. This approach is relatively recent and requires a set of optimizations for the definition of the statistical model, involving the most popular methods artificial neural networks (ANNs) [141,142]. Different researchers refer this methodology as being extremely robust and with a high predictive power, since it involves a huge set of statistical methods [141,142]. ANNs present, therefore, a great potential for the saliva analysis [143,144].

Finally, and regardless of the statistical methods used, the results obtained should be submitted to a final stage of validation to challenge the robustness of the method developed. This step is critical to evaluate the errors of the developed model and its sensitivity and specificity. Often, despite the many optional parameters involved in the different statistical tools available, default parameters are selected without proper consideration of the overfitting consequences this may have. One strategy to detect this problem more easily is to follow a double cross-validation [139,140]. This evaluation is measured by the classification rate of the model [126,139,145]. The statistical analysis is, therefore, a powerful tool for the determination of potential biomarkers, regardless of the type of analytes involved (variables), as this obviously includes ODs and the analysis of salivary VOCs.

## 7. Current Challenges and Future Perspective

Whereas the precedent section of this review explicitly elaborates the growing potential and promising future applicability of salivary volatilomics, readers should be aware of certain confounding/cofounding challenges and limitations of the same. Each omics approach involving liquid biological matrices such as blood, saliva or urine has its own potentials and weaknesses. While none of the above-mentioned biological matrices yet offered a clinically applicable volatile biomarker or marker set for routine diagnosis, all those substances, which have been discussed or referred in above sections are experimentally proposed biomarkers.

Fortunately, salivary VOCs measurement allows a non-invasive and repeated measurements without causing any additional pain/burden to subjects. In contrast to saliva, as blood sampling is invasive and painful, it might not be widely feasible towards future PoC applicability (i.e., while assuming personalized medicine), especially for a nonexpert person due to compliance issues. Especially in case of ODs, saliva acts as a proximal fluid that carries a lot of local/immediate metabolic, biochemical and/or pathophysiological information. On the other hand, presence or abundance of volatile metabolites in urine largely depend on the individual kidney function, glomerular filtration rate (GFR), bacterial flora in the urinary tract, water intake and antidiuretic hormone (ADH) activity and on the time of sampling. As the volume of urine largely depends on the intake of water and subsequent GFR, concentrations of aqueous soluble substances greatly vary within the same individual in different samples. Unlike saliva, a urinary measurement cannot be repeated whenever needed.

At present, saliva-based volatilomics also suffers from apparently unsolved problems due to ubiquitously varying confounders from the oral cavity environment. Contributions from food, beverage, smoking residuals and/or from oral microbiota are hard to differentiate. In addition, substances/metabolites with low aqueous solubility are very unlikely to pass onto saliva from the systemic circulation. Furthermore, local enzymatic (e.g., amylase, maltase, lysozyme etc.) activities within the oral cavity may give rise to many breakdown metabolites, which are hard to differentiate as exogenous (by origin) substances. As the standard secretory actions/functions of the salivary glands are still unknown, no normal range exists for saliva volume in individuals. Therefore, saliva volume and corresponding substance concentrations may vary arbitrarily in different samples from the same subject.

In order to address those issues, more systematic measures should be employed on studies concerning physiological, biochemical and/or metabolic variations of salivary volatilome in healthy population. Along with the improved analytical identification and quantification of VOCs, both sampling- and analytical standards need to be established before conducting diagnostic/screening studies on handful of subjects via applying complex statistical methods in cross-sectional mode. Advanced and hyphenated analytical applications can trace hundreds of VOCs up to low parts per trillion by volume (pptV) range, which are the independent and/or partially dependent variables (i.e., regulated by many known or unknown factors) and often employ unprecedent complexities towards reliable clinical interpretations of data. Thus, we must invest more effort on the fundamental understanding of the true origin (endogenous/exogenous, pathological/therapeutic etc.) and physiochemical properties of the volatiles in the saliva and their rational/probable clinical relationships to an oral pathology. Cross-sectionally observed differential expressions (i.e., pilot studies) in salivary volatiles must be validated/reproduced within large independent cohorts via prospective follow up. Most importantly, at such infancy of salivary omics, it is rather more important to gain basic knowledge on the effects of oral pathophysiological prognosis on VOCs compositions over the course of time.

We must realize our present analytical potentials, clinical knowledge and expertise in order to understand what is possible to achieve via salivary omics. Studies should avoid claims of early diagnosis via observed differential expressions of markers from patients at a late/advanced state of a disease. Understanding of appropriate statistical tools and interpretation of evaluated data are mandatory attributes to overcome many challenges and limitations. Finally, saliva is a dynamic matrix and can serve as an excellent phenotypic window for non-invasive monitoring of physiology, metabolism, oral pathophysiology and/or administered therapy. Salivary omics thus holds potential promise towards its future paths in relation to personalized medicine.

## Figures and Tables

**Figure 1 molecules-25-03098-f001:**
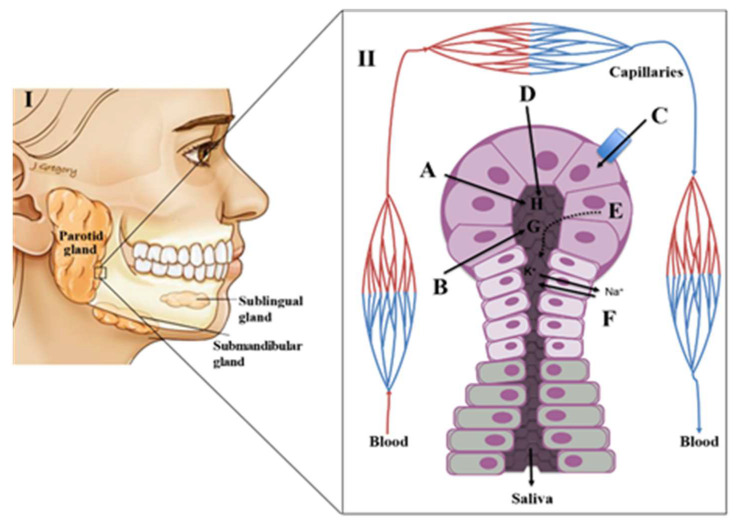
(**I**)-The major salivary glands; (**II**)–Acini structure and the different mechanisms of transport of plasma compounds into a salivary gland. A–Entry of components by simple filtration; B–Entry of liposoluble compounds by passive diffusion C–active transport; D–Active pumping of Na+ ions and concomitant entry of H2O; E–component produced/secreted by the salivary glands; F–Pumping of Na+ ions into the blood producing hypotonic fluid; G–Liposoluble compounds; H–H_2_O, electrolytes (adapted from [16]).

**Figure 2 molecules-25-03098-f002:**
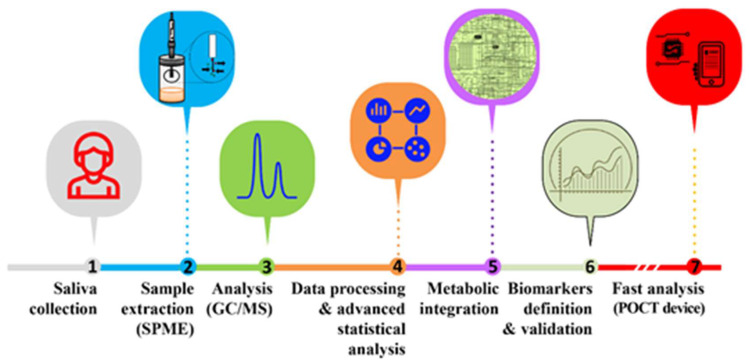
Typical experimental layout for the analysis of salivary VOCs and identification of putative volatile biomarkers for the clinical environment using POCT devices ([95,96]).

**Table 1 molecules-25-03098-t001:** Relevant salivary VOCs with potential for oral disease discrimination.

*Condition* Putative Volatile Biomarker	Metabolic Context	Ref
***Periodontal disease***		
pyridine and three methylpyridine isomers (picolines)	detected in patients but not in controls	[78]
hydrogen sulphide	oral bacteria infection	
methyl mercaptan	oral bacteria infection	
***Halitosis***		
dimethyl disulphide	oral bacterial infection	[78]
dimethyl disulphide, carbon disulphide, VSCs	drug-related metabolism	[83]
VSCs	microbial degradation products of the sulphur-containing amino acids cysteine, cystine and methionine	[78]
VSCs	augmented levels detected upon anxiety challenge	[84]
VSCs, aliphatic amines, branched chain fatty acids, indole and phenol	oral bacteria metabolism	[77]
Putrescine, cadaverine, histamine, tyramine, indole, skatole, mercaptans and sulphides	microbial metabolism of proteinaceous substrates	[85]
2,3-butanedione; 2,3-pentanedione; Phenol; pyrrole; indole and dimethyl disulphide	bacterial metabolism of lipids and carbohydrates	[80]
indole and skatole	bacterial fermentation products of tryptophan	[78]
phenol and p-cresol	bacterial putrefaction metabolites of phenolic amino acids	
***Oral candidiasis***		
3-methyl-2-butanone and styrene	*Candida albicans* infection	[86]
*p*-xylene, 2-octanone, 2-heptanone and *n*-butyl acetate	*Candida krusei* infection	
***Oral cancer***		
1,4-dichlorobenzene; 1,2-decanediol; 2,5-di-*tert*-butylphenol and *E*-3-decen-2-ol	identified in head and neck cancer cohorts	[87]
***Dietary origin***		
2-heptanone, benzaldehyde, dodecanal, 2-butyl-1-octanol, allyl isothiocyanate	examples of ketones, aldehydes, alcohols, esters and VSCs obtained from our diet	[80]
***Oxidative stress***		
hexanal and nonanal	general markers for oxidative damage (endogenously produced from membrane lipid oxidation)	[80]
***Environmental contaminants (air pollutants)***		
long-chain alkane derivatives (hexane, octane and undecane); aromatic compounds (as benzene, toluene, xylenes and styrene)	common air pollutants found in saliva	[78]

Legend: VSCs–volatile sulphur compounds.

**Table 2 molecules-25-03098-t002:** Selected examples of different experimental layouts used to characterize salivary VOCs.

Experimental Layout/Condition	Relevant VOCs Identified	Ref
HS-SPME/GC-MS		
Breast cancer	3-methyl-pentanoic acid, 4-methyl-pentanoic acid, phenol and p-tert-butyl-phenol (Portuguese samples) and acetic, propanoic, benzoic acids, 1,2-decanediol, 2-decanone, and decanal (Indian samples)	[95]
Control subjects	twenty-one VOCs detected in saliva samples, mostly aldehydes	[122]
Halitosis and Submandibular Abscesses	23 VOCs specific for halitosis and 41 for abscess	[123]
TFME-GC/MS		
OSCC	Twelve salivary VOCs were characteristic of OSCC patients	[108]
HS-trap/GC-MS		
Control subjects	34 VOCs present in all samples analysed (n = 100)	[80]
SBSE-GC/MS		
Control subjects	Excellent reproducibility for a wide range of salivary compounds, including alcohols, aldehydes, ketones, carboxylic acids, esters, amines, amides, lactones, and hydrocarbons	[81]
Control subjects	Comparison of individual and gender fingerprints using different biofluids (sweat, urine and saliva)	[111]
gas-diffusion flow injection analysis-GC/MS		
	acetaldehyde	[124]
DCM extraction and derivatization followed by GC/MS analysis		
women	2-Nonenal-ovulatory specific salivary VOCs throughout menstrual cycle	[125]

Legend: DCM–Dichloromethane, HS-trap/GC-MS–headspace-trap gas chromatography-mass spectrometry, HS-SPME/GC-MS–headspace solid phase microextraction–gas chromatography–mass spectrometry, OSCC–oral squamous cell carcinoma, TFME–Thin Film Microextraction, SBSE–stir bar sorptive extraction, VOCs–volatile organic metabolites.

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
