# Peer review of "Unravelling the Potential of Salivary Volatile Metabolites in Oral Diseases. A Review"

_molecules, 2020, doi:10.3390/molecules25133098_

Round 1

Reviewer 1 Report

The present article reviews the potential of salivary volatile metabolites in oral diseases. This review presents interesting information and it is well written and well organized. So it deserves publication after implementing some changes and some minor corrections to improve the quality of the manuscript.

My only concern related with this manuscript is that section 5 is devoted to describe the characteristics of the analytical platforms used in the volatomic analysis of saliva. However, these characteristics are general and no information relative to the determination of VOCs in saliva is given in this section. In my opinion, authors could include a table in this section detailing what VOCs were analyzed in saliva by the analytical platforms described and under what conditions (SPE, NTME, TFME, etc). As in the previous section 4, different VOCs are indicated as potential biomarkers of oral diseases, the information on their analysis could be very interesting for the reader complementing the previous information given on these compounds. At least, some particular examples could be included in section 5 and perhaps some example of the analysis of a saliva sample (figure) could also be shown. Concrete examples could also be commented in paragraph 6.

I do not see Figure 2 cited in the text.  This figure should be cited in the manuscript before it is included in the text.

Line 129, I suppose that authors mean two-dimensional gel electrophoresis (2DE). So, DE should be replaced with 2DE.

Lines 135-137, the sentence “This result reveals that the lower disease resistance offered by periodontitis patients which is reflected by the reduced antimicrobial properties exhibited by their saliva” should be revised. Something is wrong.

Line 152, when indicating the short chain fatty acids, different anions are cited but not the acids. In addition, are all these fatty acids?

Line 255, I suppose that authors mean “important candidate protein biomarkers” and not important candidate protein. This should be revised.

Line 279, this line has to be revised since the words “both local” seem not to have sense here.

Line 504, what is pptV?

Minor comments

H2O should be replaced with H2O in the caption of Figure 1.

The title of paragraph 3 (Putative salivary biomarkers for Oral diseases (ODs) should be replaced with “Putative salivary biomarkers for oral diseases” since ODs has already defined before in the text.

Line 109, “its not” should be replaced with “It is not”.

Line 113, “provably” should be replaced with “probably”.

Line 294, a space between “as” and “aliphatic” should be deleted.

Line 312, “in vitro” should be in italics.

Line 319, delete one “in” between reviewed and [74].

Line 324, other verb such as “achieved” should be used instead of “employed”.

Line 341, “and” should be replaced with “an”.

Line 475, has is correct? Or it should be replaced with have?

Line 480, replace “an” with “a”.

Author Response

The present article reviews the potential of salivary volatile metabolites in oral diseases. This review presents interesting information and it is well written and well organized. So it deserves publication after implementing some changes and some minor corrections to improve the quality of the manuscript.

Answer: we are very grateful for the valuable comments and suggestion made by the reviewer. Please find bellow our comments.

My only concern related with this manuscript is that section 5 is devoted to describe the characteristics of the analytical platforms used in the volatomic analysis of saliva. However, these characteristics are general and no information relative to the determination of VOCs in saliva is given in this section. In my opinion, authors could include a table in this section detailing what VOCs were analyzed in saliva by the analytical platforms described and under what conditions (SPE, NTME, TFME, etc). As in the previous section 4, different VOCs are indicated as potential biomarkers of oral diseases, the information on their analysis could be very interesting for the reader complementing the previous information given on these compounds. At least, some particular examples could be included in section 5 and perhaps some example of the analysis of a saliva sample (figure) could also be shown. Concrete examples could also be commented in paragraph 6.

Answer: This is a very pertinent suggestion and a new table (Table 2) was included in the manuscript to include details about the works cited in section 5. Furthermore, new references were included as well as a new subsection devoted to SBSE-related approaches for the analysis of salivary VOCs.

I do not see Figure 2 cited in the text.  This figure should be cited in the manuscript before it is included in the text.

Answer: Figure 2 was properly introduced in the text.

Line 129, I suppose that authors mean two-dimensional gel electrophoresis (2DE). So, DE should be replaced with 2DE.

Answer: This was a type and has been corrected.

Lines 135-137, the sentence “This result reveals that the lower disease resistance offered by periodontitis patients which is reflected by the reduced antimicrobial properties exhibited by their saliva” should be revised. Something is wrong.

Answer: The sentence was revised.

Line 152, when indicating the short chain fatty acids, different anions are cited but not the acids. In addition, are all these fatty acids?

Answer: Unfortunately, in the study cited, Aimetti et al (2012) was not able to provide an absolute quantification of the metabolites identified. Possibly because of this reason, the authors did not include the acids. Eventually, with an absolute quantification, further metabolites would be included as statistically significant in the discrimination of patients and controls.

Line 255, I suppose that authors mean “important candidate protein biomarkers” and not important candidate protein. This should be revised.

Answer: The sentence was revised.

Line 279, this line has to be revised since the words “both local” seem not to have sense here.

Answer: The sentence was revised.

Line 504, what is pptV?

Answer: It is parts per trillion by volume. Eventually it is not an obvious abbreviation and so it was explained in the text and added to the list of abbreviatures.

Minor comments

H2O should be replaced with H2O in the caption of Figure 1.

Answer: revised.

The title of paragraph 3 (Putative salivary biomarkers for Oral diseases (ODs) should be replaced with “Putative salivary biomarkers for oral diseases” since ODs has already defined before in the text.

Answer: revised.

Line 109, “its not” should be replaced with “It is not”.

Answer: revised.

Line 113, “provably” should be replaced with “probably”.

Answer: revised.

Line 294, a space between “as” and “aliphatic” should be deleted.

Answer: revised.

Line 312, “in vitro” should be in italics.

Answer: revised.

Line 319, delete one “in” between reviewed and [74].

Answer: revised.

Line 324, other verb such as “achieved” should be used instead of “employed”.

Answer: revised.

Line 341, “and” should be replaced with “an”.

Answer: revised.

Line 475, has is correct? Or it should be replaced with have?

Answer: It should be “have”.

Line 480, replace “an” with “a”.

Answer: revised.

Reviewer 2 Report

I carefully read the review paper submitted by Dr. Jorge A. M. Pereira et al. The main contents of this manuscript is volatile compounds while wider topics, e.g. non-volatile compounds were also reviewed. Overall, the manuscript is acceptable for the publication since it was well structured and important topics were reviewed. The following are my suggestions to improve the manuscript but not mandatory.

At the section 5.1.1. The use of SPME was introduced as one of the common tools to collect volatile compounds. It is true that this tool has been commonly for exhalation collection. The problem of this tool is the storage duration is relatively short compared to the freezing storage. Therefore, considering clinical study including a large number of cohorts, researchers have to measure the volatile compounds frequently in different batches. Quantification of metabolites, including volatile compounds, requires the rigorous quality control to eliminate unexpected batch effects. The following paper is not for volatile compounds but the concept of quality control of multiple batches for LC-MS-based quantification was described. I recommend the authors to include the topic of quality control of quantification, especially under the use of SPME and GC/MS.

  • Saigusa D et al, Establishment of Protocols for Global Metabolomics by LC-MS for Biomarker Discovery. PLoS One. 2016, 11(8):e0160555.

Most of the papers just compare the disease and control groups and the effects of various factors on the reproducibility of the potential biomarkers are not accessed. Therefore the papers including only a few samples are sometimes difficult to yield reproducible results.

One of the problems is the sample collection and storage issue. For LC/MS salivary metabolites, the following papers investigated these effects. I recommend the authors incorporate a similar topic considering volatile compounds.

  • Wang Q et al, Investigation and identification of potential biomarkers in human saliva for the early diagnosis of oral squamous cell carcinoma. Clin Chim Acta. 2014, 427:79-85.
  • Tomita A et al, Effect of storage conditions on salivary polyamines quantified via liquid chromatography-mass spectrometry. Sci Rep. 2018, 8(1):12075.

As written by the authors at citing refs. 5, 9, 10 (lines 53 - 55), the saliva is affected by internal and external factors. There are several papers to investigate the relationship between these factors and salivary metabolites. The Following papers is examples. If similar papers especially for salivary volatile compounds are available, cite these papers and discuss their effects on the salivary makers discussed in your paper.

  • Sugimoto, M. et al, Physiological and environmental parameters associated with mass spectrometry-based salivary metabolomic profiles. Metabolomics, 2013, 9(2), 454-463.
  • Shigeo Ishikawa et al, Effect of Timing of Collection of Salivary Metabolomic Biomarkers on Oral Cancer Detection, Amino Acids, 2017, 49(4):761-770.
  • Shigeo Ishikawa et al, Discrimination of Oral Squamous Cell Carcinoma From Oral Lichen Planus by Salivary Metabolomics Oral Dis, 2020 26(1):35-42.

One of the most important issues of this manuscript is how the salivary biomarkers should be biologically and clinically validate. In the second paragraph of 3.3, the authors discussed the proteomics-based salivary biomarkers of OSCC, e.g. refs. 54 and 55. However, the validation study of these markers in Taiwan is not so good reproducibility and only a few markers showed consistent aberrance between HC and OSCC. Therefore ref 59 did not validate the mathematical model developed in the refs 54 or 55 but they developed a new mathematical model using their own data collected in ref 59. This is not an actual validation. Unfortunately, no proteomics data did now show actually reproducible results, to my knowledge. The clinical validation of salivary volatile compounds is also quite important and these issues should be discussed in section 5.

Biological validation is also quite important. The ref 50 compared OSCC tissues samples and consistently elevated biomarkers in both OSCC tissues and saliva was identified. Such type of the biological and/or rational relationship between salivary biomarkers and its source should be discussed and introduced in section 5.

In the second paragraph of the 6 Data Analysis, the authors pointed out the linearity problem of the PLS-DA. I agree with the authors’ comments. In addition to the comments, the rigorous validation of PLS-DA is also required. Especially, ANN has a large number of options, such as initial weight, the number of the hidden layer, and the number of the nodes of the layers. Thus, there is a risk of overfitting. The following paper to insists on the needs of double cross-validation but many papers utilized default parameters and just provided R2 and Q2 value. The validation topic should discuss based on the salivary volatile compounds.

  • Johan A. et al, Assessment of PLSDA cross validation, Metabolomics, 2008, 4, 81–8

Author Response

I carefully read the review paper submitted by Dr. Jorge A. M. Pereira et al. The main contents of this manuscript is volatile compounds while wider topics, e.g. non-volatile compounds were also reviewed. Overall, the manuscript is acceptable for the publication since it was well structured and important topics were reviewed. The following are my suggestions to improve the manuscript but not mandatory.

Answer: we are very grateful for the valuable comments and suggestion made by the reviewer. Please find bellow our comments.

At the section 5.1.1. The use of SPME was introduced as one of the common tools to collect volatile compounds. It is true that this tool has been commonly for exhalation collection. The problem of this tool is the storage duration is relatively short compared to the freezing storage. Therefore, considering clinical study including a large number of cohorts, researchers have to measure the volatile compounds frequently in different batches. Quantification of metabolites, including volatile compounds, requires the rigorous quality control to eliminate unexpected batch effects. The following paper is not for volatile compounds but the concept of quality control of multiple batches for LC-MS-based quantification was described. I recommend the authors to include the topic of quality control of quantification, especially under the use of SPME and GC/MS.

  • Saigusa D et al, Establishment of Protocols for Global Metabolomics by LC-MS for Biomarker Discovery. PLoS One. 2016, 11(8):e0160555.

Answer: we agree that SPME is not particularly suitable for sample storage. In fact, it is advisable not store the extracted VOCs in the SPME for long periods and this is a major constraint when large cohorts have to be analysed involving automatic injection systems. In such cases, a significant delay between sample extraction and analysis may occur, therefore causing some lost of the trapped VOCs. This limitation was included and discussed in the manuscript.

Most of the papers just compare the disease and control groups and the effects of various factors on the reproducibility of the potential biomarkers are not accessed. Therefore the papers including only a few samples are sometimes difficult to yield reproducible results.

Answer: we agree that conclusions based in the comparison of a low number of samples are not robust and should be considered very carefully. For this reason, we included the information about the number of samples involved in each study cited in this review. Nevertheless, we discussed this point a little further in the manuscript.  

One of the problems is the sample collection and storage issue. For LC/MS salivary metabolites, the following papers investigated these effects. I recommend the authors incorporate a similar topic considering volatile compounds.

  • Wang Q et al, Investigation and identification of potential biomarkers in human saliva for the early diagnosis of oral squamous cell carcinoma. Clin Chim Acta. 2014, 427:79-85.
  • Tomita A et al, Effect of storage conditions on salivary polyamines quantified via liquid chromatography-mass spectrometry. Sci Rep. 2018, 8(1):12075.

Answer: we agree that this is a very relevant topic in terms of sample collection and storage for different biological fluids, including saliva, and particularly in what concerns to LC/MS analysis. However, the focus of this review is the volatomics of saliva and so a deeper revision of other omics aspects falls out of the scope of this review.

As written by the authors at citing refs. 5, 9, 10 (lines 53 - 55), the saliva is affected by internal and external factors. There are several papers to investigate the relationship between these factors and salivary metabolites. The Following papers is examples. If similar papers especially for salivary volatile compounds are available, cite these papers and discuss their effects on the salivary makers discussed in your paper.

  • Sugimoto, M. et al, Physiological and environmental parameters associated with mass spectrometry-based salivary metabolomic profiles. Metabolomics, 2013, 9(2), 454-463.
  • Shigeo Ishikawa et al, Effect of Timing of Collection of Salivary Metabolomic Biomarkers on Oral Cancer Detection, Amino Acids, 2017, 49(4):761-770.
  • Shigeo Ishikawa et al, Discrimination of Oral Squamous Cell Carcinoma From Oral Lichen Planus by Salivary Metabolomics Oral Dis, 2020 26(1):35-42.

Answer: we acknowledge the suggestion and references proposed and we have included the first two references in section 2.1. Regarding volatomics of saliva, to the best of our knowledge the interplay of internal and external factors on the saliva production and composition has not been addressed so far under this scope. The third reference suggested is related to the topic discussed in section 3 and was added to the subsection 3.3.

One of the most important issues of this manuscript is how the salivary biomarkers should be biologically and clinically validate. In the second paragraph of 3.3, the authors discussed the proteomics-based salivary biomarkers of OSCC, e.g. refs. 54 and 55. However, the validation study of these markers in Taiwan is not so good reproducibility and only a few markers showed consistent aberrance between HC and OSCC. Therefore ref 59 did not validate the mathematical model developed in the refs 54 or 55 but they developed a new mathematical model using their own data collected in ref 59. This is not an actual validation. Unfortunately, no proteomics data did now show actually reproducible results, to my knowledge. The clinical validation of salivary volatile compounds is also quite important and these issues should be discussed in section 5.

Biological validation is also quite important. The ref 50 compared OSCC tissues samples and consistently elevated biomarkers in both OSCC tissues and saliva was identified. Such type of the biological and/or rational relationship between salivary biomarkers and its source should be discussed and introduced in section 5.

Answer: the topic of biomarker validation, particularly the absence of validation in many studies proposing biomarkers for different conditions, is highlighted throughout the review. We always used the nomenclature “putative biomarkers” precisely to create the awareness in the reader that further validations of the proposed biomarkers are mandatory before any metabolite can be used as a reliable biomarker. Particularly in what concerns to salivary volatomics, which is a relatively new field, there are few, if any, volatile biomarkers identified in saliva and properly validated to be used in the clinical environment at this moment. The main goal of this manuscript is precisely to unveil this potential. Therefore, we reinforce this idea in section 3, but we consider that a deeper discussion of biomarkers validation will make this manuscript too long and descriptive.

In the second paragraph of the 6 Data Analysis, the authors pointed out the linearity problem of the PLS-DA. I agree with the authors’ comments. In addition to the comments, the rigorous validation of PLS-DA is also required. Especially, ANN has a large number of options, such as initial weight, the number of the hidden layer, and the number of the nodes of the layers. Thus, there is a risk of overfitting. The following paper to insists on the needs of double cross-validation but many papers utilized default parameters and just provided R2 and Q2 value. The validation topic should discuss based on the salivary volatile compounds.

  • Johan A. et al, Assessment of PLSDA cross validation, Metabolomics, 2008, 4, 81–8

Answer: We discussed a little further this topic in section 6 which was reviewed in detail by Westerhuis et al (2008) (this is the reference suggested and already included in the original submission – ref 132 in this revised version) and by SzymaÅ„ska et al (2012) (also previously included, ref 133 in this revised version)